# Review of Outcomes after Salvage Surgery for Recurrent Squamous Cell Carcinoma of the Head and Neck

**DOI:** 10.3390/cancers15194692

**Published:** 2023-09-23

**Authors:** Dana M. Hartl, Joanne Guerlain, Philippe Gorphe, Madan Kapre, Neeti Kapre Gupta, Nabil F. Saba, K. Thomas Robbins, Ohad Ronen, Juan P. Rodrigo, Primož Strojan, Antti A. Mäkitie, Luiz P. Kowalski, Jatin P. Shah, Alfio Ferlito

**Affiliations:** 1Department of Anesthesia, Surgery, and Interventional Radiology, Head and Neck Oncology Service, 94805 Villejuif, France; 2Department of ENT and Head Neck Surgery, Neeti Clinics Nagpur, Nagpur 440010, India; 3The Winship Cancer Institute, Emory University, Atlanta, GA 30322, USA; 4Department of Otolaryngology Head and Neck Surgery, Southern Illinois University Medical School, Springfield, IL 62703, USA; 5Department of Otolaryngology—Head and Neck Surgery, Galilee Medical Center Affiliated with Azrieli Faculty of Medicine, Bar Ilan University, Safed 5290002, Israel; 6Department of Otolaryngology, Hospital Universitario Central de Asturias, University of Oviedo, IUOPA, ISPA, CIBERONC, 33204 Oviedo, Spain; 7Institute of Oncology Ljubljana, 1000 Ljubljana, Slovenia; 8Department of Otorhinolaryngology—Head and Neck Surgery, Research Program in Systems Oncology, University of Helsinki and Helsinki University Hospital, FI-00029 Helsinki, Finland; 9Head and Neck Surgery Department, University of São Paulo Medical School, Sao Paulo 05403-000, Brazil; 10Head and Neck Surgery and Otorhinolaryngology Department, A C Camargo Cancer Center, Sao Paulo 01509-001, Brazil; 11Department of Surgery, Head and Neck Service, Memorial Sloan Kettering Cancer Center, New York, NY 10065, USA; 12Coordinator of the International Head and Neck Scientific Group, 35100 Padua, Italy

**Keywords:** head and neck cancer, squamous cell, chemoradiation, salvage surgery

## Abstract

**Simple Summary:**

Local recurrences, isolated regional recurrences, and loco-regional recurrences are frequent after treatment for advanced stage head and neck squamous cell carcinoma. Salvage surgery may be possible in selected patients, taking into account the location and stage of the recurrent disease, patient age and comorbidities, tumor HPV status, and functional sequelae in relation to initial therapy. Results vary according to these parameters. Adjuvant treatments after salvage surgery are under investigation. The aim of this review is to present current knowledge concerning the incidence and management of recurrent head and neck squamous cell carcinoma and current data concerning survival and morbidity after salvage surgery.

**Abstract:**

Surgery with adjuvant chemoradiotherapy or chemoradiotherapy is the mainstay in treatment for advanced stage head and neck squamous cell carcinoma; however, locoregional recurrences are frequent. Salvage surgery could be proposed in selected patients to improve local control, disease-free, and overall survival. Factors for improved disease-free and overall survival in patients treated with salvage surgery include age, tumor location, the initial T stage, HPV status, resection margins, and the time elapsing from the initial treatment. Clinical trials with adjuvant therapies have shown promise after salvage surgery in terms of tolerance and response, but clinical guidelines for using these adjuvant treatments are currently lacking. The aim of this review is to present current knowledge concerning the incidence and management of recurrent head and neck squamous cell carcinoma and current data concerning survival and morbidity after salvage surgery.

## 1. Introduction

Surgery with postoperative chemoradiation or definitive chemoradiation is the main first-line treatment modality for advanced-stage head and neck squamous cell carcinoma (HNSCC) [1,2]. Locoregional control rates vary with disease stage and with differing anatomic tumor sites, with higher rates of local recurrence in patients that have oral cavity carcinoma, followed by laryngeal and hypopharyngeal carcinomas. HPV-related oropharyngeal carcinoma shows a lower rate of recurrence compared to non-HPV-related oropharyngeal carcinoma [3]. In locally advanced HNSCC, roughly half of patients experience a recurrence, primarily locally and/or regionally [4].

Salvage surgery includes open, transoral, and robot-assisted techniques. When feasible, salvage surgery has been shown in retrospective studies to improve local control, disease-free, and overall survival (DFS and OS) in patients treated with salvage surgery compared to patients either not eligible for salvage surgery or treated non-surgically. Five-year overall survival ranges from 21% to 61% [5,6,7,8]. Several factors, such as patient age, tumor site, and initial tumor stage, the time lapse between initial treatment and the recurrence, and the association between regional and local recurrence, modulate the oncologic results of salvage surgery [5,9,10,11]. These factors should be considered when selecting patients for salvage surgery [12]. Postoperative reirradiation, when feasible in patients with adverse features, has demonstrated improved locoregional control but significant toxicity [13,14]. Adjuvant immunotherapy has shown promising results in recent clinical trials and is under further investigation. In the following paragraphs, we explore current data on salvage surgery (taking into account relevant publications as of the year 2000) in patients with locoregional failure after initial definitive treatment for advanced squamous cell carcinoma and its relevance in the evolving era of other treatment modalities.

## 2. Definition of Recurrent Disease after Chemoradiation or Surgery and Chemoradiation

Recurrent disease is generally defined as the re-appearance of cancer after a lapse in time during which there is no detectable disease; however, there is no uniform definition of locally recurrent HNSCC. This general definition distinguishes recurrence from persistent disease in which there is no point at which the tumor is no longer detectable or a short period where the tumor is not detectable or evaluable. The time period without a detectable tumor used to define recurrence varies among authors and publications, and there is no consensus today as to the disease-free period needed to define recurrence as opposed to persistent disease or second primary tumors. In many publications, the cutoff used to distinguish recurrence from persistent disease is 6 months [5,6,8]. Others define “early” recurrences as those arising less than 12 months from initial therapy and “late” recurrences as those appearing more than 5 years after initial treatment [15]. However, recurrences in the same anatomic site occurring 5 years after initial treatment are most often classified and treated as metachronous new primary tumors [16].

Recently, a definition has been proposed by a team of specialists based on a systematic review of the literature [17]. This definition—termed the Odense-Birmingham definition—states that a locally recurring cancer must be in the same anatomical subsite or in an adjacent subsite located less than 3 cm from the primary lesion, the time of appearance of the lesion must be within 3 years from the initial therapy, and, for oropharyngeal lesions, the new lesion must have the same HPV status as the initial tumor.

Recurrence can occur in the anatomical site of the initial disease, in neck lymph nodes, and/or in distant metastatic sites. Metachronous primary disease occurs after a lapse of time during which there is no detectable disease but occurs in an anatomical site that is distinct from that of the initial primary tumor. This distinction might be impossible to determine when recurrence occurs in a site close to or adjacent to the initial primary tumor site. 

## 3. Diagnosis of Recurrent Disease

Recurrent disease may be suspected upon physical examination, through clinical signs and symptoms, a follow-up with cross-sectional imaging or ^18^FDG-PET-CT imaging. Clinical symptoms are non-specific and may be misleading. Patients may experience chronic pain after definitive radiation therapy or chemoradiation, which might be at the forefront of the symptomatology and mask pain related to a recurrent tumor. Radiation-induced trismus may also mask trismus due to the tumor and make the physical examination of the oral cavity and oropharynx almost impossible. The re-appearance of referred otalgia on the side of the initial tumor may be relevant for the suspicion of recurrence. Dysphagia and pharyngeal and/or laryngeal oedema may also be chronic and lead to the delayed diagnosis of recurrence. Laryngeal immobility may be observed after definitive radiation therapy or chemoradiation but may also be an indirect sign of tumor recurrence. Mucosal ulcerations or chronic infection with radionecrosis might present in a clinically similar manner to recurrent or persistent cancer, and differential diagnosis is often difficult as well.

Lee et al. investigated optimal surveillance visit schedules for patients who had survived advanced head and neck cancer [18]. They retrospectively analyzed recurrence and second primary events in 673 patients over 11 years at two tertiary care medical centers. Their analysis suggests that surveillance regimens for HPV-positive oropharyngeal cancers and for nasopharyngeal cancers could be significantly reduced, while those for laryngeal carcinoma and HPV-negative oropharyngeal cancers could remain approximately the same as current guidelines. Surveillance regimens for hypopharyngeal cancers, however, should be intensified. Circulating tumor DNA (ctDNA) has been proposed as a biomarker to detect minimal residual disease and monitor early molecular-level recurrence in patients with head and neck cancer, leading to personalized ctDNA assays for therapy planning and follow-up [19]. However, the clinical impact of surveillance using ctDNA requires further validation.

In a case-matched study comparing the histopathology of upfront versus salvage total laryngectomies by Zbären et al., [20] recurrent tumors were more likely to be multifocal, poorly limited, and more poorly differentiated than primary laryngeal tumors. Due to these histopathologic specificities and pre-existent post-radiation fibrosis and edema, the interpretation of cross-sectional and metabolic imaging might be difficult and misleading. Inflammation and radiation-induced necrosis may show a false positive contrast enhancement on cross-sectional imaging and/or false positive lesions on metabolic imaging. It is, therefore, important to ensure the precise timing of ordering metabolic imaging for enhanced diagnostic accuracy, such as scans performed after 10–12 weeks of completed treatment, which have much higher specificity than those conducted earlier [21]. The high negative predictive value of computed tomography and ^18^FDG-PET-CT for nodal metastasis was recently reported by Rovira et al. in a group of 22 recurrent/persistent primary head and neck cancer patients after bioradiotherapy who underwent salvage surgery, suggesting the abstention of treatment of the neck in radiologically negative neck cases [22].

When performing biopsies, one must target the lesions visualized both clinically and on imaging, and often, deep biopsies must be performed to identify submucosal recurrence. Repeated biopsies are frequently necessary for negative initial biopsies due to edema and necrosis. Performing the directed biopsies of PET-CT-detected subsites might further reduce the chances of a false negative biopsy [23].

## 4. Salvage Surgery: Incidence

Approximately 50% of patients with initially advanced stage III-IV disease treated with radiation therapy or chemoradiation present with a local and/or regional recurrence at some point during follow-up. In the prospective study by Beitler et al., the 2-year locoregional recurrence-free survival rates ranged from 45.7 to 53.8% for the four different radiation therapy fractionation arms delivered to a total of 1076 patients with stage III/IV tumors with all anatomic locations combined [4]. In a meta-analysis of 101 clinical trials with a total of 18,951 patients, over 70% of patients experienced a cancer-related event during follow-up [24].

Not all patients are eligible for salvage surgery. Thirty-seven percent of patients with a laryngeal or hypopharyngeal tumor were eligible for salvage surgery in the study by Van der Putten et al., [25] which corresponded to a reported range of between thirty-three and sixty-six percent in other studies [5,26,27,28,29,30]. Salvage surgery has been shown to be performed more often in younger patients (<60 years old) and in those with laryngeal primary tumors [25,29]. For de Ridder et al., salvage surgery was performed in 53% (104) of 198 patients with recurrence and was significantly more often performed in patients with recurrent laryngeal cancer, isolated nodal recurrence, and HPV-related cancers [30]. According to a recent meta-analysis of 25 studies relating to salvage surgery, the factors significantly associated with worse overall survival were being age 70 years old in age, initial stage IV disease, a disease-free interval of less than 12 months, and associated regional recurrence [31]. Perineural invasion on salvage pathology was also identified as a predictor of poor outcomes in 241 surgically salvaged patients reported by Akali et al. [32].

## 5. Salvage Surgery: Survival

In patients eligible for and treated with salvage surgery, two-year overall survival rates were reported to range from 27 to 71% [6,8,10,33] and five-year rates from 21% to 67% [8,34,35,36]. These rates are higher than those of patients with recurrence treated without salvage surgery, keeping in mind that candidate patients treated with salvage surgery are selected patients [3,30,33,36]. In the study by Patil et al., of the 113 patients deemed eligible for salvage surgery, 91 accepted surgeries, while 22 did not [7]. Of the 91 patients who accepted surgery, only 78 actually underwent salvage surgery, with a median survival of 22.0 months compared to 9.7 months in patients who were unwilling to undergo salvage surgery. In the study by de Ridder et al., among the 104 patients treated with salvage surgery, after a median follow-up of 10 months (range 1–96 months), 18% were alive with no evidence of disease, whereas 48% had died either from unsuccessful salvage or from a new disease recurrence [30]. The interpretation of these results is complex due to the retrospective nature of the majority of these studies, which introduces selection bias, the heterogeneity of the patient cohorts in terms of tumor location, their stage at recurrence, patient age and comorbidities, and the inherent variability in performing surgical procedures [37].

Several factors have, thus, been shown to be implicated in disease-free and overall survival after salvage surgery: -Initial tumor stage: A lower initial tumor stage carries a more favorable outcome; the initial N3 stage has been shown to be a factor for lower overall survival after salvage surgery [10].-The stage of the recurrent tumor [8].-Patient age: patients aged < 58–65 years [9,11,33] when salvage surgery is performed have been shown in some studies to have better survival rates; however, other studies have not found age to be a significant risk factor for survival outcomes [25].-HPV-positive disease: patients with HPV-positive disease have higher rates of disease-free survival after salvage surgery [30].-Resection margins after salvage surgery: high rates of R1 resection—14 to 40%—after salvage surgery have been reported by experienced teams despite wide, macroscopically complete resection [5,34]. An R1 resection has been shown in several studies to be an independent factor for poorer overall and disease-free survival [5]. However, in the study by Leeman et al., margin status or the presence of an extranodal extension after salvage surgery for oral cavity carcinoma was not shown to affect overall survival [3].-The time elapsed between initial therapy and salvage surgery: late recurrences, after 1 year [5,10,11,15] or 2 years [9,33], have been shown in some studies to have more favorable outcomes after salvage surgery.-Tumor location: disease-free survival after salvage surgery for laryngeal primary tumors has been shown to be significantly better than hypopharyngeal and oropharyngeal tumors [8,9,34]. Higher overall survival rates have been reported in patients treated for isolated lymph node recurrences or isolated local recurrences without lymph node recurrence compared to simultaneous local and regional recurrences [5,6,8].-Perineural invasion in the salvage resection specimen has also been implicated as a factor of less favorable prognosis in terms of survival for laryngeal carcinoma [38,39].

An initial tumor designation of stage IV and the presence of a concomitant local and regional recurrence were found to be factors that decreased overall survival. For patients without either of these factors, 2-year overall survival was 83% versus 49% for patients with one of these prognostic factors and 0% in patients with both of these factors [5]. Likewise, Hamoir et al. identified three prognostic factors—an initial tumor stage of III or IV, a combined local and regional recurrence, and a non-laryngeal tumor location [8]. Patients with none of these factors had a 2-year disease-free survival rate of 96%, while those with one risk factor had a 2-year disease-free survival rate of 62%; those with two risk factors had a 2-year DFS of 35%, and those with all three risk factors had a 2-year DFS of 29%. Applying the same prediction model to 577 patients undergoing salvage surgery, Quer et al. reported a 5-year disease-specific survival for those with zero, one, two, or three predictors of 82.2%, 47.2%, 29.5%, and 20.2%, respectively [40]. More recently, in 164 patients with recurrent head and neck cancer reported by Lupato et al., salvage surgery resulted in progression-free survival and overall survival of 36.6% and 44.2%, respectively, at 5 years. Four factors were found to be independently associated with poor prognosis: age over 70 years, initial stage IV, a time to recurrence of less than 12 months, and loco-regional recurrence. Five-year overall survival in patients with 3–4 unfavorable factors versus 0–1 unfavorable factors were 0% versus 65.7%, respectively [41]. Kim et al. studied risk factors for patient mortality during the first year after salvage surgery [42]. Of the 191 patients studied, 53 (27.7%) died within 1 year. The independent risk factors for death within 1 year included the Charlson Age-Comorbidity Index, an initial T3 or T4 class tumor, and a disease-free interval of less than 6 months.

For recurrent nasopharyngeal carcinoma, in a retrospective study of 272 patients, 30.9% received surgery, whereas 35.7% received re-irradiation only [43]. The 5-year overall survival for the whole group was 30.2%, 56.3% for the surgery group, and 21.8% for the re-irradiation group. Older age and advanced rT classification were adverse prognostic factors, whereas surgery was associated with a more favorable survival outcome. In a meta-analysis of 17 studies and of the 779 patients treated for recurrent nasopharyngeal carcinoma, the 5-year overall survival and local recurrence-free survival rates were 51.2% and 63.4%. The 5-year overall survival rate was 63% in patients receiving surgery and adjuvant radiotherapy compared to 39% in patients receiving surgery alone (*p* = 0.05) [44]. Independent predictors of the outcome on the multivariate analysis included sex, N classification, surgical approach (endoscopic versus open), adjuvant treatment, and margin status. Both endoscopic surgery and reirradiation were independent predictors of improved survival. 

Reported 3-year survival rates following salvage surgery for oropharyngeal carcinoma ranged from 34 to 62% [45,46]. In the study by Zafereo et al., twenty-six out of thirty-nine patients (67%) developed a second recurrence after salvage surgery for oropharyngeal cancer [47]. The 3-year overall survival rate for patients who underwent salvage surgery or received re-irradiation, palliative chemotherapy, or supportive care were 48.7%, 31.6%, 3.7%, and 5.1%, respectively. For patients who underwent salvage surgery, an older age (*p* = 0.03), the absence of a disease-free interval (*p* < 0.01), and advanced recurrent tumor stage (*p* = 0.07) were associated with lower overall survival. Patients with recurrent neck disease (*p* = 0.01) and positive surgical margins (*p* = 0.04) experienced higher rates of recurrence after salvage surgery. For HPV-related oropharyngeal cancer, Taniguchi et al. found heterogeneity when reporting overall and disease-specific survival outcomes after salvage surgery [45]. Two-year overall survival, specifically after salvage surgery for HPV-related oropharyngeal carcinoma, was reported at 51% in one study [48] and 91% in another [49], and 5-year overall survival was reported at 27% in one study [48] and 43% in another [50]. These factors are summarized in Table 1.

For laryngeal carcinoma, Scharpf et al. reported a series of 147 patients who underwent salvage laryngectomy for recurrent or persistent disease after radiotherapy or chemoradiation [51]. The 2-year locoregional failure rate was 21.8%, and the overall survival was 65%. Regarding multivariable analysis, a sarcomatoid/spindle cell pathology, lymphovascular invasion, and advanced tumor stage were associated with lower disease-free survival. In a meta-analysis of salvage surgery for laryngeal carcinoma, including 235 patients undergoing salvage transoral laser microsurgery after primary radiation therapy or chemoradiation, the local control rates at 1, 3, and 5 years were 74.2%, 53.9%, and 39.1%, respectively [52]. Disease-specific survival at 1, 3, and 5 years was 88.4%, 67.8%, and 58.9%, respectively. Meulemans et al. reported the results of a retrospective multicenter cohort of 405 patients who underwent salvage total laryngectomy for residual, recurrent, or second primary carcinoma of the larynx or hypopharynx after initial radiation therapy or chemoradiation [38]. Five-year overall survival, disease-specific survival, disease-free survival, and locoregional relapse-free survival were 47.7%, 68.7%, 42.1%, and 44.3%, respectively. In a multivariable model, increasing tumor stage at salvage, an increasing number of metastatic nodes, hypopharyngeal and supraglottic tumors, positive resection margins, and perineural invasion were independent negative prognostic variables. Finally, Shapira et al. performed a meta-analysis on 15 retrospective studies, including 323 patients, to evaluate outcomes after salvage partial laryngectomy for recurrence [53]. Total laryngectomy-free survival was 90.4% after open salvage partial laryngectomy and 78.6% following transoral surgery in these selected patients. Oncologic outcomes are summarized in Table 2.

## 6. Salvage Surgery: Elective Neck Dissection

Unlike the lymphatic spreading to the neck of treatment-naïve tumors of the head and neck that follow a predictable pattern, isolated local recurrences pose a dilemma as to whether to surgically treat the neck and to what extent due to distorted lymphatic drainage. 

An aberrant lymphatic spread in patients with a recurrence of oral cavity cancer has been demonstrated using a sentinel lymph node biopsy. den Toom et al. performed a sentinel lymph node biopsy in patients who had been treated previously for nodal disease. Unexpected drainage patterns were observed in 30% of patients, and in 12% of patients, no lymphatic drainage was seen [54]. While a level IV neck involvement in oral tongue-naïve patients could be expected in only 2.8%, the authors found level IV to be involved in 4/13 (31%) of patients who experienced lymphatic drainage at this unexpected neck level [54,55].

Three meta-analyses of elective neck dissection in rN0 necks failed to show a survival advantage for neck dissection over observations in recurrent laryngeal cancer after salvage total laryngectomy [56,57,58]. For laryngeal carcinoma, the incidence of occult metastasis in dissected necks ranged from 9 to 16% overall but was 17.8% for supraglottic tumors [56]. For Gross et al., in recurrent supraglottic carcinoma, an advanced T classification of the initial tumor carried a higher risk of positive neck disease [57]. Similarly, in a study by Mazerolle et al. on 239 patients who underwent salvage surgery for pharyngeal or laryngeal tumors with no evidence of lymph node involvement (cN0), occult lymph node metastases were found in 9% of patients, though the authors found no difference in disease-specific survival between the group of 143 patients undergoing prophylactic neck dissection and the 96 patients with a neck observation only [9]. In a retrospective multicenter study of patients who underwent salvage total laryngectomy, Dassé et al. observed a 9.1% overall rate of occult nodal metastases in cN0 patients and a lower rate of occult metastases in patients with recurrences staged rT1-rT2 and recurring after a period of 12 months following initial therapy. No survival difference was observed in patients treated with elective neck dissection or observation [59].

## 7. Salvage Surgery: Morbidity

Postoperative complications, reported in 22–92% of cases, are more frequent after salvage surgery and include early complications such as a fistula, bleeding, infection, flap necrosis, aspiration pneumonia, the decompensation of comorbidities, and postoperative death [5]. Late complications include permanent tracheostomy, permanent gastrostomy, and spinal accessory nerve dysfunctions. In the group of 234 patients in the study by Locatello et al., the 30-day postoperative complication rate after salvage surgery was 44.4%, with half of these complications classified as Clavien–Dindo grade ≥ 3 [60]. In the study by Hamoir et al. on multivariate analysis, the occurrence of a major complication was found to be correlated with a decrease in disease-free survival [8]. In the study by Tan et al., 63% of patients presented with postoperative complications: early complications in 21%, late complications in 24%, both early and late complications in 18% of patients, and postoperative death in 8% of patients [5]. Richey et al. also reported an early complication rate of 24% [10]. Hamoir et al. reported a fistula rate of 16.5%, hemorrhage of 4.5%, flap failure requiring reintervention in 8.3%, and major medical complications in 23% [8]. In the meta-analysis of 16 studies with 729 patients reported by Elbers et al., the fistula rate from pooled data was 33%, with 24% of patients experiencing wound infections and 3% a flap failure [35].

In a systematic review, Van der Putten et al. reported fistula rates ranging from 4 to 73% after salvage laryngectomy [25]. In a recent study by Šifrer et al., in 110 patients treated with salvage total laryngectomy, the incidence of a pharyngocutaneous fistula was 32.7%, with surgical wound infections and the dose of radiotherapy identified as independent risk factors for fistula formation [61]. In retrospective studies, the reported rate of the use of vascularized flaps is over 50% [5,29,62,63,64]. The use of a vascularized flap– either pectoralis or free flap—was shown in a retrospective study from seven centers, including 359 patients who reduced their fistula rate in salvage laryngectomy [65]. An additional systematic review and meta-analysis has shown that the use of vascularized tissue after salvage laryngectomy reduces the risk of a fistula by 40% [66]. Finally, a retrospective study conducted by the Microvascular Committee of the American Academy of Otolaryngology-Head & Neck Surgery, including 486 patients from 33 centers, found that the use of a vascularized flap significantly reduced the fistula rate in salvage surgery for hypopharyngeal cancers [67].

In terms of functional results, in the study by Tan et al., 53% of patients treated with salvage surgery did not require a feeding tube or gastrostomy, and 55% did not have a tracheotomy or tracheostoma [5]. These functional results were statistically better than those of patients treated palliatively. A recent meta-analysis of 34 articles revealed a long-term feeding tube and tracheostomy tube rates of 18% and 7%, with a long-term feeding tube frequency dependent on the type of surgical procedure: 41% after open oral and oropharyngeal surgeries, 25% after transoral robotic procedures, 11% after total laryngectomy and 4% after partial laryngectomies [68]. Functional outcomes were shown in a meta-analysis by Goodwin et al. to be related to the anatomic location of the tumor and to the stage at the time of salvage surgery [69]. Outcomes in terms of swallowing and diet were significantly better for laryngeal tumors. Patients with stage I or II tumors had better outcomes in terms of oral feeding, speech, and quality of life. In the study conducted by the Microvascular Committee of the American Academy of Otolaryngology-Head & Neck Surgery, reconstruction with a muscular flap during the salvage surgery of the larynx and/or hypopharynx led to worse 12-month speech and oral feeding scores compared to reconstruction without muscle [67].

For eligible patients, functional outcomes are improved by the use of transoral laser or robot-assisted techniques for salvage surgery, when possible, compared to open salvage surgery [70,71]. However, there is clearly a selection bias in patients who are found to be suitable for trans oral surgery. In the study by White et al., 64 patients who underwent salvage TORS for recurrent oropharyngeal carcinoma were matched by the TNM stage with 64 patients who underwent open salvage resection. Patients treated with TORS were found to have a significantly lower incidence of the tracheostomy and feeding tube, shorter overall hospital stays, decreased operative time and blood, and a decreased incidence of positive margins. The 2-year recurrence-free survival rate was higher in the TORS group than in the open-surgery group (74% and 43%, respectively) (*p* = 0.01) [70]. Again, after interpreting these results, one needs to keep in mind the selection bias for the TORS group.

## 8. Salvage Surgery: Adjuvant Treatments

Current guidelines recommend enrolling patients in a clinical trial when feasible [72].

### 8.1. Re-Irradiation after Salvage Surgery

For patients treated with re-irradiation after salvage surgery, two-year overall survival rates range from 21% to 81%, and 2-year locoregional control rates from 4% to 100%. Grade 3 or 4 late toxicities are reported in 13–50% of patients, with the rate of treatment-related deaths ranging from 0% to 8% [73].

A phase III clinical trial including 130 patients evaluated adjuvant re-irradiation in terms of disease-free survival and overall survival. Patients treated with salvage surgery for recurrence or for a metachronous second primary head and neck cancer in an already-irradiated field were randomized between re-irradiation (with split course radiotherapy delivering 60 Gy over 11 weeks with concomitant 5-fluoro-uracil and hydroxyurea) versus the best supportive care [13]. Loco-regional failure was the major cause of death, and an increase in both acute and late toxicities was observed in the re-irradiation arm. At 2 years, 39% of patients in the adjuvant re-irradiation arm and 10% in the supportive care arm experienced grade 3 or 4 late toxicity according to Radiation Therapy Oncology Group criteria (*p* = 0.06). Disease-free survival was significantly improved in the re-irradiation arm, with a hazard ratio of 1.68 (95% CI, 1.13 to 2.50; *p* = 0.01), but overall survival was not statistically different between these two patient groups. A second randomized study of post-salvage surgery reirradiation was compared with two modalities: 53 patients were randomized to receive either 60 Gy over 11 weeks (once-daily split course) with concomitant 5FU—hydroxyurea or 60 Gy (1.2 Gy twice daily)—over 5 weeks with cetuximab. The main endpoint was acute toxicity, which did not differ between the treatment arms, nor did the overall or disease-free survival rates [14].

The risk factors for survival after reirradiation were evaluated in a retrospective database study on 103 patients, of which 46 also underwent salvage surgery [74]. The authors found that survival was significantly related to pre-existing organ dysfunction and comorbidities, as well as to the interval of time from the previous irradiation, the tumor volume at the time of reirradiation, and the dose used for reirradiation. In the group of patients with both pre-existing organ dysfunctions and comorbidities, reirradiation only had a palliative role. In a retrospective study of 21 patients undergoing salvage neck dissection, a 50% 5-year overall survival rate was observed in those patients selected to undergo adjuvant interstitial high-dose brachytherapy as well [75]. Thus, patients with no comorbidities, no functional morbidity from the previous irradiation, and who are more than 6 months from the previous irradiation might be better candidates for reirradiation in an attempt to improve disease-free survival, providing that complete wound healing from salvage surgery has been obtained before re-irradiation [13].

### 8.2. Adjuvant Chemotherapy after Salvage Surgery

A recently published phase II trial evaluated weekly oral methotrexate and oral celecoxib twice daily for 6 months [76]. The primary endpoint was disease-free survival in patients with recurrent head and neck cancer who had undergone salvage surgery and were deemed ineligible for adjuvant re-irradiation. One hundred and five patients were randomized to receive treatment or an observation. After a median follow-up of 30 months, no difference in survival was found between the study arms.

### 8.3. Adjuvant Immunotherapy after Salvage Surgery

Following the important breakthrough showing improved survival with nivolumab for recurrent and/or metastatic platinum-refractory disease [77,78], a recent phase II study has shown that post-salvage surgery adjuvant nivolumab is well tolerated, with a 2-year disease-free survival rate of 71% and a 2-year overall survival rate of 73% in 39 patients [79]. This study showed no difference in terms of disease-free survival between PD-L1 positive and PD-L1 negative patients. Another recent phase II study of 28 patients treated with pre- and post-salvage surgery nivolumab and lirilumab showed a low toxicity profile and a pathologic response to the surgical specimen in 43% of cases, with a major response in 14%. The two-year disease-free survival and overall survival in patients showing a pathologic response was 64% and 80%, respectively [80]. There are no studies to date, however, that compare adjuvant immunotherapy to a placebo. A phase II study is underway to investigate the outcomes and tolerance of adjuvant nivolumab or nivoloumab plus ipilimumab in comparison with a historic cohort treated with re-irradiation [81]. Another phase II trial is underway to investigate pembrolizumab, either alone or with adjuvant radiation therapy, compared to adjuvant chemoradiation after salvage surgery for recurrent or second primary head and neck cancers [82].

### 8.4. Other Treatments 

Other treatments that might be associated with salvage surgery are theoretical or have only been reported in small cohorts of patients. One theoretical approach that needs to be validated in clinical trials is the administration of immunotherapy before salvage surgery to induce a response that could improve outcomes and/or optimize the selection of patients eligible for salvage surgery [83]. The role of boron neutron capture therapy in the adjuvant setting after salvage surgery has not yet been investigated but has been employed as a sole modality in the recurrent setting [84], as has photodynamic therapy [85,86].

## 9. Conclusions

Salvage surgery is an important therapeutic option in selected patients, with better disease-free survival outcomes in patients < 65 years old with lower-stage recurrences occurring more than 6 months from the initial therapy, laryngeal recurrences, and isolated local or regional disease (versus both local and regional disease). Flap placement (free or pedicle) has been shown to improve the postoperative course. Prospective studies are currently aiming to improve survival with adjuvant reirradation, chemotherapy, and/or immunotherapy regimens after salvage surgery.

## Figures and Tables

**Table 1 cancers-15-04692-t001:** Factors implicated in improved outcomes after salvage surgery [5,8,9,10,30,34,35,40,45,46].

Factor	
Patient age	<58–65 years
Initial and recurrent tumor stage	Lower stages generally have more favorable outcomes
HPV-positive disease	Oropharyngeal carcinomas
Tumor location	Laryngeal primaries (glottic, in particular)
Nodal and/or local recurrence	Isolated nodal or local recurrence are more favorable than combined local and nodal recurrence
Time elapsed from initial radiation therapy	>1 or >2 years
Resection margins	R0 resection (contradicting studies)
Perineural invasion	The absence of perineural invasion is more favorable

**Table 2 cancers-15-04692-t002:** Outcomes of salvage surgery.

Initial Tumor Location	Salvage	Outcomes [References]
**Nasopharynx**	Surgery alone	5-year OS*: 56.3% [43]
	Surgery alone	5-year OS: 39% [44]
	Surgery + re-irradiation	5-year OS: 63% [44]
**Oropharynx**	Surgery	3-year OS: 48.7% [47]
	Re-irradiation alone	3-year OS: 31.6% [47]
**Oropharynx HPV+**	Surgery	5-year OS: 27–43% [48,50]
**Larynx**	Salvage total laryngectomy	2-year OS: 65% [51] 5-year OS: 47.7% [38]
	Salvage transoral surgery	5-year DSS*: 50.9% [52]

OS*: overall survival. DSS*: disease-specific survival.

## Data Availability

Data sources are available by contacting the corresponding author.

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
