# Peer review of "Review of Outcomes after Salvage Surgery for Recurrent Squamous Cell Carcinoma of the Head and Neck"

_cancers, 2023, doi:10.3390/cancers15194692_

Round 1
Reviewer 1 Report
I read with interest the paper entitled Review of Outcomes of Salvage Surgery for Recurrent Squamous Cell Carcinoma of the Head and Neck by Hartl and coworkers.
This a comprehensive narrative review regarding the outcomes of salvage surgery for recurrent HNSCCs.
There is a similar paper that has been recently published by the same journal
(https://www.mdpi.com/2072-6694/15/9/2625).
I feel that this paper provide to the reader an updated state-of-art review focused on the outcomes of salvage treatment, being either salvage surgery or a combination of salvage surgery with other adjuvant treatment.
It can be considered a valuable addition to the scientific literature on the topic and it might be a useful starting point for other research on the same topic.
Author Response
The authors thank the reviewer for their time and effort in reviewing our manuscript and for the positive comments.
Reviewer 2 Report
A comprehensive updated review about the oncological results of salvage surgery in HNC, from a panel of recognized experts in the field. Not much to add but:
1) please specify how the articles were selected (time span?)
2) please add a table divided by tumoral subsite regarding the outcomes and potential salvage strategies according to the latest evidence in the literature
Author Response
Response to reviewer 2
We thank the reviewer for their time and effort in reviewing our manuscript and for their very constructive comments. We have made the suggested modifcations highlighted in yellow:
- On page 3 we have added the time slot when the cited papers were selected
- On page 11 we have added a table summarizing the data according to anatomic subsites.
We hope that the manuscript will be found to be suitable for publication.